# Key Determinants Influencing Treatment Decision-Making for and Adherence to Active Surveillance for Prostate Cancer: A Systematic Review

**DOI:** 10.3390/jpm15070315

**Published:** 2025-07-15

**Authors:** Pani Nasseri, Jorien Veldwijk, Christa Niehot, Esmee F. H. Mulder, Esther W. de Bekker-Grob, Monique J. Roobol, Lionne D. F. Venderbos

**Affiliations:** 1Department of Urology, Erasmus University Medical Center Rotterdam, 3015 GD Rotterdam, The Netherlands; c.niehot@erasmusmc.nl (C.N.); e.f.h.mulder@erasmusmc.nl (E.F.H.M.); m.roobol@erasmusmc.nl (M.J.R.); l.venderbos@erasmusmc.nl (L.D.F.V.); 2Erasmus School of Health Policy & Management, Erasmus University Rotterdam, P.O. Box 1738, 3000 DR Rotterdam, The Netherlands; veldwijk@eshpm.eur.nl (J.V.); debekker-grob@eshpm.eur.nl (E.W.d.B.-G.)

**Keywords:** prostate cancer, active surveillance, treatment decision, treatment choice adherence, discontinuation

## Abstract

**Background/Objectives**: Men choosing active surveillance (AS) for low- and intermediate risk prostate cancer (PCa) must weigh its harms and benefits against those of active treatment (AT). To understand factors influencing treatment decision-making (TDM) for and adherence to AS, we performed a systematic review. **Methods**: This systematic review followed the PRISMA guidelines and was registered with PROSPERO (ID CRD42024490427). A comprehensive search strategy from 1990 to 2024 was executed across multiple databases, including Medline and Embase. Studies were included if they examined factors influencing TDM for AS or active monitoring (AM) and adherence to AS/AM. **Results**: Of the 8316 articles identified, 223 articles were eligible for inclusion. The decision for AS was influenced by different factors, including comprehensive information about treatment options, social support, and wanting to avoid side-effects. Key reasons to choose AT over AS included a lack of information from healthcare professionals about AS and fear of disease progression. Reasons for adhering to the AS protocol included better quality of life and social support. While non-adherence to the AS protocol is prompted by, e.g., biopsy burden and uncertainty, AS discontinuation is generally a results of medical reasons (namely disease upgrading) or from anxiety and family pressure. **Conclusions**: Numerous factors influence men’s treatment pathway choices. Involving family members in shared decision-making and ensuring that patients have detailed information about AS as a treatment option could help to improve AS uptake. Addressing psychosocial challenges through education and family involvement could improve AS adherence. These insights can help healthcare providers by addressing men’s needs during TDM and AS.

## 1. Introduction

For many years, the primary treatments for clinically detected localized prostate cancer (PCa) were radical prostatectomy (RP) and radiotherapy (RT) [1]. However, due to the evolution of diagnostic methods, the early detection of PCa has become possible, offering a greater chance of cure. These early detection activities, however, have also resulted in a considerable stage and grade shift, with up to 42% of the tumors identified considered low-risk [2]. If undetected, these tumors would likely not have progressed to clinical or physical symptoms for men at any time in their lives. Detecting these low-risk, so-called indolent tumors is also referred to as overdiagnosis [2]. Since low-risk tumors carry a minimal risk for patient mortality and providing treatment to these men would most likely not improve their health outcomes [3], treating these tumors with RP or RT is considered overtreatment, impacting both the quality of life (QoL) of patients and incurring additional costs [4,5,6]. An alternative treatment option for patients with low-risk PCa and increasingly also for patients with intermediate-risk PCa, defined as Gleason 3 + 4, is active surveillance (AS) [7]. Instead of active treatment (AT), AS monitors patients using a combination of PSA testing, digital rectal examination (DRE), prostate biopsies, and magnetic resonance imaging (MRI) [8]. Following AS monitoring protocols can help patients to either delay or avoid radical treatments, thereby decreasing overtreatment.

Once a man has opted for AS, adherence to the AS monitoring protocol is of utmost importance, not only for PCa-related outcomes [9,10], but also for two other reasons. First, adhering to the monitoring protocol can function as a psychological safety net [11], as it involves follow-up in specialized hospitals and perceived consistency in medical information, which are associated with reduced anxiety and distress [12]. Second, AS provides better QoL and health-related quality of life (HRQoL) than radical treatments, which cause side effects not encountered by patients receiving AS [13,14,15]. Opting for AS still provides men with the ability to switch to AT, which could lead to the discontinuation of AS. Approximately 50% of patients discontinue AS within 5 years. Reasons for discontinuation vary from medical reasons (e.g., disease progression (21.7–41%) or physician’s recommendation) to personal reasons (e.g., anxiety (5–15%) and social influences) [16,17,18,19]. While discontinuation due to medical reasons is generally not classified as non-adherence, discontinuation linked to personal factors such as anxiety or social influences is considered a form of non-adherence.

More men with low- and intermediate-risk PCa are choosing AS as their primary treatment [20,21]. This decision is challenging because it involves weighing the risks of cancer progression against the risks of AT [22]. Factors influencing this choice include patient preferences, anxiety, knowledge, health outcomes, and physician recommendations [22,23].

Although AS has played a significant role in reducing overtreatment [24] and provides a relatively high QoL [25], adherence to the scheduled prostate biopsies during AS declines steadily over the course of the monitoring protocol from 81% at 1-year follow-up to only 33% at 10 years [9]. Understanding the reasons behind AS acceptance and adherence is needed to develop targeted interventions to facilitate adherence and informed decision-making. To address this need, this study aims to identify factors influencing the decision for AS after low- and intermediate-risk PCa diagnosis, as well as factors affecting (non)adherence to the AS monitoring protocol

## 2. Materials and Methods

This systematic review followed the Preferred Reporting Items for Systematic Reviews and Meta-Analysis (PRISMA) guidelines [26]. The review protocol was registered with PROSPERO (ID CRD42024490427).

### 2.1. Study Eligibility

This systematic review used the results of previous work by Kinsella et al. [27] and McIntosh et al. [28] as guidance whilst developing a new search.

Studies were considered eligible for inclusion in the final analysis if they investigated factors that influenced the decision-making process for AS or active monitoring (AM), as well as adherence to the AS protocols. Furthermore, studies that reported on and/or compared QoL and decision-making between AS and other treatments were included. Only studies available in English were eligible for inclusion in the analysis.

Studies reporting on men undergoing “watchful waiting” (WW) were included only after verifying that the respondents were actually engaged in AS rather than WW, due to occasional confusion between the two. The following definition of WW was used for this verification: WW involves providing no initial treatment and monitoring the patient with the intention of providing palliative treatment if there is evidence of disease progression [29]. Therefore, studies that combined AS/WW groups were also excluded.

### 2.2. Search Strategy

An exhaustive search strategy was developed by an information specialist (C.D.N.) in cooperation with the lead author (P.N.). The search was developed in Embase.com, optimized for sensitivity, and then translated to other databases. The search was carried out in the databases Medline ALL via Ovid, Embase.com, the Web of Science Core Collection (Science Citation Index Expanded; Social Sciences Citation Index; Arts and Humanities Citation Index; Conference Proceedings Citation Index—Science; Conference Proceedings Citation Index—Social Science and Humanities (1990–present); and Emerging Sources Citation Index), and the Cochrane Central Register of Controlled Trials via Wiley. Additionally a search was performed in Google Scholar from which the 200 most highest-ranked references were downloaded using the software Publish or Perish [30].

The search strategies for Medline and Embase used relevant thesaurus terms from Medical Subject Headings (MeSH) and Emtree, respectively. In all databases, terms were searched in titles, abstracts, and author keywords. The search contained terms for (1) prostate cancer (2) adherence and (3) active surveillance. The complete search strategies for all databases are available in the (Appendix A). No limits were used in the search strategy. No study registries were searched, but Cochrane CENTRAL retrieves contents from ClinicalTrials.gov and World Health Organization’s International Clinical Trials Registry Platform (ICTRP). No authors or subject experts were contacted, and we did not use unindexed journals in the field.

### 2.3. Study Selection and Data Extraction

The references were imported into EndNote and duplicates were removed by the information specialist (C.D.N.).

Two reviewers (P.N. and L.D.F.V.) independently screened titles and abstracts in Covidence. Any discrepancies in inclusion or exclusion decisions were resolved by discussion with a third reviewer (J.V.). Full texts that were potentially eligible were then reviewed and screened by one author (P.N.). Two samples of each 10% were reviewed and screened by L.D.F.V. and J.V to avoid discrepancies.

The full texts of abstracts published between 2020 and 2024 were manually searched. Abstracts outside this range were excluded from the final review.

A data extraction form was developed, based on the initial review of the first few articles. The form was reviewed and revised by J.V., E.d.B.G., L.D.F.V. and M.R. Before being used for data extraction, the extraction form was pilot tested by P.N. on the first few full texts and adjusted when needed (addition of themes). Data extracted included publication year, authors, journal name, title, study design, population, data collection method, and outcomes.

### 2.4. Quality Appraisal

Quality appraisal of the included literature was conducted using the mixed methods appraisal tool (MMAT) for systematic mixed studies review [31]. Three researchers (P.N., E.F.H.M. and L.D.F.V.), each conducted evaluations separately. Any differences in assessment were sorted out through agreement with a fourth reviewer (J.V.). Papers were evaluated based on five criteria, with each criterion contributing 20% to the overall quality score. A higher score indicated better quality, with the maximum possible score being 100%.

## 3. Results

The literature search identified 8316 citations which were screened for title/abstract. A total of 7623 citations were excluded because they were commentaries, on another topic, duplicates, or included a WW population. The articles included in Kinsella et al. [27] and McIntosh et al. [28] that were not already in our search were added manually, leaving 704 citations for full-text screening. Of these, 481 citations were excluded, which resulted in 223 papers ultimately being included. Details of the full-text exclusions can be found in Figure 1.

### 3.1. Quality Appraisal

Using the MMAT, the included papers were evaluated against five criteria related to their specific research design. Of the included studies, 91 received a full quality score (100%), 83 scored 80%, 30 scored 60%, 15 scored 40%, and 4 studies met only 20% of the quality criteria (Table 1). The most commonly identified methodological issues were the absence of confounder adjustment, missing baseline measurements, and incomplete outcome data.

This review has been divided into three parts: decision-making for AS, adherence to AS, and discontinuation of AS. These have been further categorized into themes found during data extraction (Table 2): clinical factors (e.g., healthcare organization, healthcare provider, and cancer characteristics), patient factors (e.g., comorbidities, side-effects, knowledge/information, and QALY) and social factors (e.g., social influences, social support, and social acceptance) (Figure 2).

### 3.2. Decision-Making

#### 3.2.1. Clinical Factors

Three cohort studies [32,116,134] showed that consultations at a **multidisciplinary clinic** were associated with more men choosing AS. In addition, Bellardita et al. [48] and Hurwitz et al. [116] found that men attending multidisciplinary clinics benefit from the comprehensive information obtained at the consultations. Receiving treatment at an academic medical center is associated with a higher likelihood of choosing AS [158]. Men in rural areas with no academic hospitals were less likely to decide for AS in comparison to men living in urban areas [112,147].

Patients who indicated that their treatment decision was made by the **physician** or those who engaged in shared decision-making (SDM) were more likely to choose AS [43,109]. While having conversations with a physician influences the decision-making for AS, Fitch et al. [94] found that it sometimes adds to a patient’s distress. Different labels for cancer, during these conversations, can cause emotional reactions that do not align with the actual medical risk. Berlin et al. [53] showed that by avoiding terms such as cancer and adenocarcinoma, the probability of preferring AS increases from 75 to 82% with patients and 65–82% for partners. Men who regard their cancer as ‘low-grade’ or ‘slow-growing’ are more likely to select AS.

A total of 25% of the studies showed that a **physician’s recommendation** is a principal factor in men’s decision-making [51,74,75,76,79,97,99,144,164,173,176,177,194,226]. Patients were more likely to choose AS if seen by a urologist or oncologist compared to other physicians [67,109,119]. Davison et al. [74] found that although the majority of the men want more involvement in their treatment decision-making, the personal treatment preferences of their physician remain essential in men’s decision process [90]. This preference of physicians towards AS plays a crucial role, as some physicians who do not favor AS may not recommend it and, consequently, some men do not have access to it, also known as physician bias [99,111,173,196,226,244].

The **time between diagnosis and treatment decision** is an important factor, as some studies indicate that extending the time between diagnosis and the treatment decision increases the likelihood of opting for AS [150,164,173]. This additional time allows for obtaining comprehensive information and being in touch with healthcare professionals [173].

**Cancer characteristics** is another key factor in deciding to undergo AS. A total of 9% of the studies showed that the slow growth of a tumor and tumor volume in particular were mentioned as determinants influencing men’s decisions for AS [52,152,226]. However, some men prefer an aggressive approach with the removal of the cancer rather than leaving it in their bodies [22,36,150,226].

#### 3.2.2. Patient Factors

Five studies have indicated that older men (≥70) are more likely to choose AS [34,59,76,144,177]. Several studies showed that higher levels of education and the presence of comorbidities are also associated with the decision for AS [32,106,147,158,177]. However, some other studies found that higher comorbidity burden is associated with the choice for AT [42,116,130,191]. Although spirituality and religion were less frequently mentioned, they guide the decision-making process by reducing anxiety, making the choice for AS easier [100,159,185]. However, Mishra et al. [164] found that the choice for AS is not always easy since there is a notable lack of acceptance among men towards early AS.

A total of 8% of the studies reported that men wanting to avoid the **side-effects** of AT have a strong preference for AS [47,59,74,75,76,90,99,142,150,151,182,196]. Risk of urinary problems and loss of sexual function were mentioned often, as this impacts their lives and intimate relationships and may even decrease their sense of masculinity [33,74,77,100]. Bowel-related issues were mentioned less frequently but were also reported as a side-effect that men want to prevent [51,57,138]. Multiple studies showed that **fear of disease progression** during AS is a principal determinant of deciding for AT [83,111,159,244,246]. Men admit that they cannot tolerate the uncertainty of daily life without knowing whether the cancer has progressed or not [160]. Fear of prostate biopsies and the associated burden also influences men to opt for AT [33,36,164,173].

Having **complete information** about their disease, treatments, and mortality risk increases the likelihood of deciding for AS [33,44,48,76,94,145]. However, due to limited time available during consultations, it is difficult for physicians to provide all the necessary information [145]. Fitch et al. [93] found that men find the conversations about diagnosis important to obtain a clear understanding of treatment options available and hence to ensure an informed treatment decision. Three qualitative studies found that men often seek information from multiple sources such as the internet, physicians, friends, and family [93,142,150]. Men who use a decision aid in their decision-making process are more likely to opt for AS [137].

#### 3.2.3. Social Factors

**Social influences**, from partners or family members, have a big impact on the decision for treatment [74,75,99,111,164,244]. Filson et al. [92] suggest that men with partners are more likely to choose AS. Acceptance of AS by family members provides men with support, facilitating their decision to choose AS as a treatment option [99,142]. However, sometimes men feel they have to defend their decision for AS to their family since AS is often misunderstood by patients and families as withholding treatment [97,152,174,226,244]. Furthermore, men are more likely to opt for AT if their friends or family have had negative experiences with AS [36,150,160,164,188,211].

### 3.3. Adherence

#### 3.3.1. Clinical Factors

Multiple studies examined the relationship between **cancer characteristics** and adherence to AS. The Michigan Urological Surgery Improvement Collaborative registry (MUSIC) demonstrated that patients with early reassuring results were more likely to remain on AS compared to those with non-reassuring results [125].

Evans et al. [88] found that men who are diagnosed at a private hospital are more likely to adhere to AS during follow-up than those diagnosed in public hospitals. Multiple qualitative studies have shown the **importance of a urologist** during AS follow-up, with trust in and support from a urologist particularly facilitating men’s adherence to AS [173,196,228].

Chen et al. [65] found that men who saw a urologist during follow-up were more likely to receive guideline-recommended AS compared to those who did not. On the other hand, Clements et al. [68] found that appointments with advanced practice providers, including physician assistants and nurse practitioners, are more likely to be canceled. Some studies indicated that men wish for more comprehensive information from their healthcare providers during follow-up, including information on PCa, long-term outcomes of AS, procedures (biopsy), and lifestyle options [76,97,145,164].

#### 3.3.2. Patient Factors

Certain studies found that younger men (<65 years) and those with higher education levels experience greater difficulties with the prospect of long-term AS [44,140,146,173]. Conversely, Evans et al. [88] found that men aged 66 years or older were less likely to adhere to AS and that adherence was also lower among men diagnosed via the transurethral resection of the prostate (TURP) or transperineal (TP) biopsy.

The follow-up protocol during AS provides men with **reassurance** by enabling them to monitor disease progression [52,151,171]. Mader et al. [151] indicated that the continuity provided by AS established a sense of control in men, leading to reduced anxiety. Although some studies have indicated that patients experience anxiety in the days leading up to the test, this anxiety rapidly reduces upon receiving the results [52,151,171,196]. Some studies showed that anxiety and distress levels [196,214,216] during AS are within normal levels, for both patients and their partners [12,37,197].

A total of 31% of the studies have highlighted the **burden of prostate biopsies**, which men perceive as uncomfortable and increases anxiety [9,33,56,79,152,164]. Bokhorst et al. [9] provided a detailed example of the decline in biopsy adherence over time, reporting a reduction from 81% at 1 year to 33% at 10 years follow-up. Another study of Bokhorst et al. [56] showed that men with a previous complication at their biopsy were less likely to undergo another biopsy while on AS. Chen et al. [65] demonstrated that adherence to PSA, biopsy, and DRE was met by less than 50% of the patients within two years. Kalapara et al. [122] examined adherence rates and observed a decline in adherence to annual and the Prostate cancer Research Internation Active Surveillance (PRIAS)-based rebiopsy, from 92% and 89% at year 1 to 66% and 71% at year 7, respectively. Olsson et al. [175] found a decrease in biopsy rates with comorbidities and increasing age.

A total of 32% of the studies showed that following the AS protocol preserves the **QoL** of the men [196,214,216]. Kinsella et al. [18] reported that self-help strategies, including dietary changes and exercise, contributed to the QALY of men during AS and helped them feel better. Adhering to AS allowed men to maintain their normal daily routines, minimizing disruptions that could have occurred with AT [151]. The Hormonal therapy, Active Surveillance, Radiation, Operation, Watchful Waiting (HAROW) study [39] reported better HRQoL outcomes compared to that of men receiving RP. The Japanese arm of the PRIAS study [105] found better HRQoL in Japanese men on AS compared to the general population. Other studies, including Venderbos et al. [25] and Jeldres et al. [120] found similar results where men on AS report better HRQoL than men who underwent AT. Additionally, a long-term QoL follow-up study by Venderbos et al. [25] showed that the QoL of men on AS is comparable to that of men without PCa.

Several other studies reported that the HRQoL in men undergoing AS is higher than that of men receiving AT [64,107,129]. Many studies have demonstrated that men who choose AS experience fewer or less severe side-effects compared to those undergoing AT [41,42,64,80,102,107,120,166,216,222].

#### 3.3.3. Social Factors

A total of 9% of the studies have emphasized the role of strong **social support** during AS, as it enhances confidence and helps them initiate lifestyle changes [151,196,244]. Despite this, several other studies have highlighted the lack of psychosocial support for men during AS follow-up, even with men’s need for help in coping with PCa [40,44]. However, Baba et al. [40] demonstrated that the need for psychosocial support is relatively similar between patients undergoing AS and those receiving AT. A few studies noted the importance of spousal support, as spouses often guide men in making necessary lifestyle and dietary changes [128,151,174]. Yet some studies pointed out that the **pressure** men receive from their family to switch to AT while being on AS is difficult [44,164,244]. Seiler et al. [197] even showed slightly higher anxiety among partners of PCa patients. Engaging with other men diagnosed with PCa helps patients to adhere to AS [151]. Davison et al. [75] showed that a support group for men on AS was identified as important. However, Kazer et al. [128] reported that the men who participated in support groups did not find them as useful.

### 3.4. Discontinuation

#### 3.4.1. Clinical Factors

Numerous studies have shown that **cancer progression** is the primary reason for discontinuing AS, as outlined in the protocol, with progression indicated by higher Gleason scores, increased tumor volume, and elevated PSA levels [16,19,85,130,140,146,162,179,196,199,204,208,219,224]. Other studies also showed that men switch to AT due to Gleason and biopsy **upgrading** [105,132,153,175,190,215]. Fitch et al. [93] demonstrated that disease status is a crucial factor in the pursuit of AS and emphasized that changes in test results indicating disease progression were a reason to discuss discontinuing AS.

A total of 8% of the studies have found that the **presence of comorbidities** is a factor leading to the discontinuation of AS, even though it is not mentioned as often [19,196]. Kelly et al. [130] found that prostate biopsy complications due to comorbidities is one of the reasons for discontinuation in the cohort. Loeb et al. [146] showed that men with comorbidities are less likely to discontinue AS due to preferences. Lokman et al. [148] found that men on AS transitioned to WW in the presence of other health conditions, and some patients died during AS due to unrelated causes. Non-prostate cancer death was also found in other studies [16,105,153].

Four studies have shown that one of the primary reasons men switch to alternative treatments is the **recommendation of their doctor** [9,19,51,52]. Kelly et al. [130] showed that the recommendation is often due to disease progression. Lai et al. [135] found that at 3 years after diagnosis only 64.3% of men followed up by their radiation oncologist remained on AS, compared with higher percentages (75.8%, 79.1%, and 79.5%) of men followed-up by their urologist, oncologist, and primary care physician. Timilshina et al. [208] also found that treatment by a radiation oncologist is associated with a higher likelihood of AS discontinuation compared to seeing a urologist during follow-up. The Swedish National Prostate Cancer Register study [146] showed that men diagnosed in 2004 and 2005 had a lower risk of discontinuation compared to men diagnosed in the previous year 2003.

#### 3.4.2. Patient Factors

In seven of the included studies it was found that, in certain cases, patients themselves wish to switch to AT despite the absence of clinical evidence supporting the need for such a change [16,51,85,140,146,153,224]. Kelly et al. [130] found indecision regarding treatment choice as the primary reason for discontinuation. Lang et al. [140] found that patients with college education discontinue AS significantly earlier than patients with lower education levels.

A total of 7% of the studies identified that **anxiety** is a contributing factor for men discontinuing AS, although only a small number of men switch to AT due to this [19,148,153,199,204,224]. The uncertainty experienced by men is primarily due to the fear of cancer progression, which often leads to the discontinuation of AS [19,37,224]. McIntosh et al. [19] identified this theme of discontinuation as fear, worry, and uncertainty. Kinsella et al. [18] showed that men who ceased AS described their experience of AS as stressful. However, Seaman et al. [196] found that no one mentioned anxiety as a reason to discontinue AS and start AT. Repetto et al. [190] found that decision regret is a contributing factor for some patients discontinuing AS.

#### 3.4.3. Social Factors

Three studies highlighted that discontinuation is also influenced by **familyencouragement** to switch to AT [19,44,132]. Families often regard AS as an absence of treatment and consequently urge their relatives to opt for AT [44,164,244]. Three studies mention spousal encouragement instead of general family encouragement in influencing discontinuation of AS [44,140,174].

## 4. Discussion

The aim of this systematic review was to identify factors that influence low-risk prostate cancer patients when choosing for AS, as well as factors influencing (non)adherence during AS. The decision for AS was influenced mainly by physicians’ recommendations, social support, and wanting to avoid side effects. Reasons for adherence to AS were mostly driven by a good relationship with and trust in the healthcare provider and social support, while non-adherence was prompted from lack of social support and uncertainty. While discontinuation typically resulted from medical reasons as prescribed in the AS protocol, personal factors such as anxiety and family encouragement could also contribute to patients deciding to discontinue AS.

Our review showed that the role of the physician in the decision-making process of AS was among the most important factors. Therefore, physicians’ preferences can influence treatment decisions, with those who are not in favor of AS being less likely to recommend it. This is also referred to as physician bias [99,111,173,226]. Raising awareness regarding their influence on patient decision-making could help reduce such biases and improve shared decision-making. The variation in the uptake of AS is not only different between physicians but also within individual practices. A retrospective analysis performed by Cooperberg et al. [247] found that the uptake of AS in the United States varied from 4% to 78% at urology practice level (e.g., including other healthcare professionals) and from 0% to 100% at the individual urology practitioner level. This suggests that the use of AS can differ significantly between countries, regions, and possibly even within urology practices. Our findings align with these observations: we found for instance that treatment decisions in academic hospitals were more likely to favor AS compared to those in rural settings [112,147,158].

Furthermore, our systematic review revealed that trust in a urologist and direct consultation with a urologist during follow-up care, compared to consultations with other healthcare providers, such as advanced practice providers ((APP), e.g., nurse practitioners and physician assistants), contributes to AS adherence. However, this difference was relatively small, as APPs only showed a minor increase in no-show appointments [68]. In addition, other studies have highlighted the role of trust and communication with healthcare providers, regardless of their type, in optimizing adherence to AS. A qualitative study by Chen et al. [248] found that urologists observed a preference among men to discuss follow-up care with their primary care providers, which may reflect a higher level of trust in these providers for such discussions; however, this could be attributed to the fragmented nature of the American healthcare system, where the role of primary care providers may vary across settings, influencing patient preferences. These findings emphasize the importance of establishing trust-based relationships and ensuring clear communication between patients and healthcare providers to maintain ongoing engagement [176,226,248,249]. Moreover, trust and effective SDM are likely to further enhance adherence throughout AS.

In our review, we highlighted the importance of social support for patients undergoing AS, particularly in addressing the psychosocial challenges they face. We emphasized that family involvement can play a key role in improving adherence. A scoping review by Donachie et al. [250] aligns with these findings, presenting various interventions aimed at reducing the psychosocial burden experienced by PCa patients in AS. The review suggests that such support can be effectively delivered through a network of family members and friends, who can help manage the anxiety and uncertainty. Integrating family support into the care of PCa patients undergoing AS could have benefits, not only in terms of improving adherence but also in addressing the broader emotional and psychosocial challenges.

Additionally, our review found that men’s participation in support groups is a particularly beneficial strategy to reduce anxiety. It allows men to engage in open conversations, connect with peers who share similar experiences, and receive emotional support. This, in turn, has been linked to improved QoL during AS [251]. Our study further highlighted that engaging with other men diagnosed with PCa enhances adherence to AS. This finding is supported by a Report of a Movember International Consensus Meeting by Moore et al. [252], which emphasizes the role that support groups play in promoting AS adherence.

### 4.1. Clinical Implications

Findings from this review highlight several strategies for improving adherence to AS. First, clinicians should consider prioritizing clear risk communication, framing AS as a proactive, evidence-based management approach. The use of decision aids, easy language, and communication training within institutions or departments, such as team meetings on communication practices, can help clarify risk, reduce decisional regret, and ensure alignment with patient needs. Second, involving family members or even close friends in SDM from the time of diagnosis may enhance patients’ confidence and provide ongoing emotional support, particularly during follow-up appointments. Clinicians should not only address the patient but also encourage patients to involve a companion, as such individuals often play a key role in decision-making and long-term support. Third, integrating psychosocial resources into AS, such as counseling or structured support groups, can help to address common challenges such as anxiety and uncertainty. Peer-to-peer interaction can be highly impactful; men often benefit from interacting with others who are also undergoing AS or have experience with PCa. These approaches could potentially be effectively embedded within hospitals or clinics to ensure a more consistent and supportive care environment. The effectiveness and sustainability of such resources in AS should be investigated in future studies.

### 4.2. Strengths and Limitations

This systematic review provides a comprehensive overview of factors influencing treatment decision-making (TDM) and adherence to AS for PCa. By including studies that span over 30 years and addressing patient perspectives, our review offers insights into the complexity and psychosocial challenges during a men’s pathway on AS. Additionally, the large number of included studies (223) and the strong methodology (e.g., independent researchers for coding, inclusions, data extraction, and quality assessment) enhances the validity and strength of our findings.

The studies included in this review were heterogeneous, addressing different research questions and using different research methods. While this can offer a broad perspective, it also poses a limitation in terms of integrating findings, as the varied approaches may result in inconsistent outcomes. To address this, we used the MMAT quality appraisal tool with different study design checklists to ensure that we measured the quality of each study appropriately. Despite the heterogeneity of the studies, the majority nonetheless achieved a high MMAT score. Another limitation is that majority of the studies were conducted in Western countries (USA, UK, Canada, and Australia), which have distinct healthcare systems. As a result, the findings may not be easily generalizable to countries or regions with different healthcare structures. However, these regions can still draw lessons from the processes described in the literature and adapt them in a way that fits their own healthcare systems. Another limitation of this study is the lack of standardization in defining ‘adherence’ across existing research, which complicates comparisons between studies. To address this, we have used the following definition of adherence in AS from the start: the extent to which patients consistently follow the AS schedule over the course of the treatment. Another limitation is the potential for publication bias, as studies with positive findings are more likely to be published. To minimize this risk, we conducted a comprehensive and broad search strategy, within multiple databases from 1946 to present, to include a broad range of studies regardless of their outcomes.

## 5. Conclusions

Numerous factors influence men’s pathway from treatment decision-making for AS to adherence throughout AS. Despite international, national, and regional differences in healthcare systems and clinical practices, as well as the fact that this review includes evidence from studies with diverse research designs, participant groups, and recruitment, several key factors have consistently emerged. Physician recommendation, social support, and clear, comprehensive information strongly influence the initial decision to pursue AS. Insights gained from multidisciplinary clinics and academic centers with higher adherence to an AS protocol could contribute to improving AS protocols. Involving family members in SDM and ensuring that men have detailed information about AS as a treatment option could improve AS uptake. Addressing psychosocial challenges through education and family involvement could help improve the long-term adherence of AS.

## Figures and Tables

**Figure 1 jpm-15-00315-f001:**
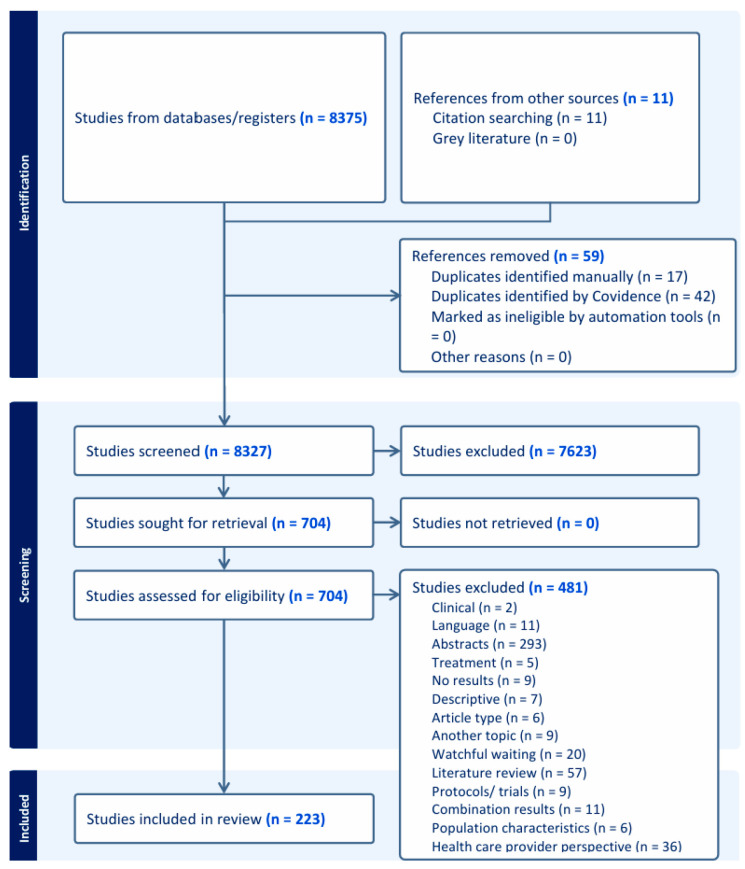
PRISMA flowchart.

**Figure 2 jpm-15-00315-f002:**
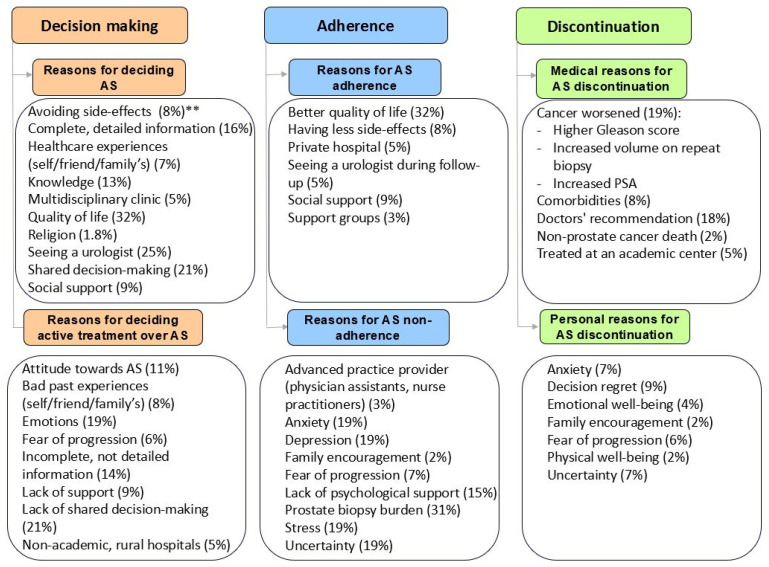
Factors influencing an active surveillance (AS) pathway in alphabetical order ** The percentages represent the proportion of studies reporting each factor. These percentages do not total 100%, as individual studies often addressed multiple factors.

**Table 1 jpm-15-00315-t001:** Overview of the included studies in the review on factors affecting decision-making for and adherence to active surveillance in men with low-/ and intermediate-risk prostate cancer.

Author and Year	Country	Study Design	Study Population	Age (Mean)	MMAT (Quality Score)
Aizer et al., 2013 [32]	USA	Retrospective cohort study	Patients with low-risk PCa	MC: 63 IP: 61	*****
Al Hussein Al Awamlh, 2023 [33]	USA	Qualitative interview study	PCa survivors who underwent AS, RP, or RT	n/a	*****
Al Hussein Al Awamlh et al., 2020 [34]	USA	Retrospective cohort registry study	Men with localized PCa	Conservative treatment: 65.12 Definitive treatment: 63.13	*****
Alvisi et al., 2020 [35]	Italy	Prospective cohort survey study	AS patients	n/a	****
Anandadas et al., 2011 [36]	UK	Prospective survey study	Patients with low-risk PCa	AS: 64.8 BR: 62.0 CRT: 64.3 S: 62.5	****
Anderson et al., 2014 [37]	Australia	Prospective cohort survey study	AS patients	65.7	**
Anderson et al., 2022 [38]	USA	Cross-sectional analysis	Men with localized PCa	n/a	****
Ansmann et al., 2018 [39]	Germany	Prospective observational survey study	AS and RP patients	AS: 67.7 RP: 63.9	****
Baba et al., 2021 [40]	Germany	Observational, cross-sectional survey study	AS, RP, biochemical relapse, and metastasized disease patients	68.2	****
Banerji et al., 2017 [41]	USA	Prospective survey study	AS and RT patients	AS: 64 EBRT: 65	*****
Barocas et al., 2017 [42]	USA	Prospective, population-based cohort survey study	RT, S, and AS patients	63.8	****
Basak et al., 2022 [43]	USA	Population-based prospective cohort survey study	Patients with low-risk PCa	n/a	**
Beckmann et al., 2021 [44]	UK	Qualitative interview study	Men who discontinued AS for AT	64	*****
Beckmann et al., 2022 [45]	UK	Patient and public involvement Delphi study	AS patients and healthcare providers	n/a	***
Beckmann et al., 2019 [46]	Sweden	Retrospective register-based study	Men with low-risk PCa	63.9	****
Bellardita et al., 2013 [47]	Italy	Prospective cohort study	AS patients	67	****
Bellardita et al., 2019 [48]	Italy	Prospective survey study	AS patients	64.8	*****
Bergengren et al., 2018 [49]	Sweden	Nationwide population-based survey study	AS, RP and RT patients	63	****
Bergengren et al., 2022 [50]	Sweden	Nationwide population-based survey study	AS, RP, and RT patients	n/a	****
Bergengren et al., 2019 [51]	Sweden	Nationwide population-based survey study	AS, RP, and RT patients	63	*****
Berger et al., 2014 [52]	USA	Mixed methods study	Men who discontinued AS and AS patients	n/a	**
Berlin et al., 2023 [53]	Canada	Discrete choice experiment study	Men with low-risk PCa and healthy men and partners	n/a	***
Berry et al., 2021 [54]	USA	Randomized controlled trial	Men with localized PCa	n/a	*
Boberg et al., 2003 [55]	USA	Quantitative cross-sectional survey study	Men with localized PCa	66	***
Bokhorst et al., 2015 [9]	The Netherlands	Retrospective registry study	AS patients	n/a	****
Bokhorst et al., 2016 [56]	The Netherlands	Retrospective cohort chart review	AS patients	n/a	****
Bosco et al., 2012 [57]	USA	Randomized controlled trial	Men with localized PCa	n/a	****
Braun et al., 2014 [58]	USA	Prospective survey study	AS patients	n/a	****
Broughman et al., 2018 [59]	USA	Population-based cohort survey study	Men with low-risk PCa	n/a	***
Burnet et al., 2007 [60]	UK	Cross-sectional survey study	AS patients	67.12	**
Burns et al., 2019 [61]	USA	Prospective cohort survey study	AS patients and healthy men	Biopsy: 61.1 No biopsy: 62.8	****
Butler et al., 2020 [62]	USA	Retrospective registry study	AS and WW patients	n/a	****
Carlsson et al., 2021 [63]	USA	Pilot study with shared medical appointments	AS patients	n/a	****
Chen et al., 2017 [64]	USA	Population-based prospective cohort	AS, RP, EBRT, and BR patients	AS: 67 EBRT: 67 BR: 66 RP: 62	***
Chen et al., 2021 [65]	USA	Prospective population-based cohort	AS patients	Adherent: 66.5 Non-adherent: 66.3	*
Chien et al., 2014 [66]	Taiwan	Quantitative longitudinal study	Men with localized PCa	70	****
Chu et al., 2015 [67]	USA	Retrospective cohort study	AS patients	61.8	****
Clements et al., 2021 [68]	USA	Retrospective review	AS patients	Patients: 69 Urologist: 66	****
Couper et al., 2009 [69]	Australia	Observational prospective survey study	Men with localized PCa	66.15	****
Crump et al., 2021 [70]	Multi-country	Retrospective study	AS patients	n/a	*****
Cuypers et al., 2018 [71]	The Netherlands	Prospective survey study	Men with PCa	66.5	***
Cuypers et al., 2018 [72]	The Netherlands	Randomized controlled trial	Men with PCa	65.3	*****
Daubenmier et al., 2006 [73]	USA	Randomized controlled trial	AS patients	Control: 66.5 Intervention: 64.8	**
Davison & Breckon, 2012 [74]	Canada	Cross-sectional survey study	AS patients	67.2	****
Davison & Goldenberg, 2011 [75]	Canada	Cross-sectional descriptive survey study	AS patients	64.49	***
Davison & Breckon, 2012 [76]	Canada	Cross-sectional survey study	Men with PCa	63.5	****
de Bekker-Grob et al., 2013 [77]	The Netherlands	Discrete choice experiment	Men with low-risk PCa and urologists	Patients: 73 Urologist: 49	****
Donachie et al., 2022 [78]	The Netherlands	A modified Delphi study	AS patients	n/a	***
Donachie et al., 2020 [79]	The Netherlands	Qualitative interview study	AS patients	67	*****
Donovan et al., 2016 [80]	UK	Prospective cohort survey study	RT, surgery, and AM patients	62	***
Dordoni et al., 2021 [81]	Italy	Longitudinal survey study	AS patients	n/a	****
Dordoni et al., 2022 [82]	Italy, Netherlands	Prospective survey study	AS patients	n/a	*****
Egger et al., 2018 [83]	Australia	Population-based prospective cohort study	AS and WW patients	n/a	*****
El-Haouly et al., 2020 [84]	Canada	Cross-sectional observational study	Men with localized PCa	68.34	*****
Eredics et al., 2017 [85]	Austria	Prospective registry study	AS patients	64	****
Erim et al., 2020 [86]	USA	Cross-sectional analysis of secondary data from a prospective cohort study	Men with PCa	64.6	****
Ernstmann et al., 2012 [87]	Germany	Longitudinal survey study	AS, WW, RP, HT, and RT patients	WW: 73.36 AS: 68.02 RP: 65.3 HT: 74.03 RT: 69.55	**
Evans et al., 2018 [88]	Australia	Retrospective cohort study	AS patients	n/a	*****
Eymech et al., 2022 [89]	UK	Qualitative interview study	AS patients	66	*****
Feldman-Stewart et al., 2001 [90]	Canada	Cross-sectional survey study	Men with low-risk PCa	66	***
Feldman-Stewart et al., 2018 [91]	Canada	Cross-sectional survey study	Men with PCa	65.7	****
Filson et al., 2021 [92]	USA	Randomized controlled trial	Men with low- and intermediate-risk PCa	69	*
Fitch et al., 2020 [93]	Canada	Qualitative comparative study	Men with PCa and healthcare providers	n/a	*
Fitch et al., 2017 [94]	Canada	Qualitative focus group study	Men with low-risk PCa	n/a	*****
Formica et al., 2017 [95]	USA	Cross-sectional study	Men with low-risk PCa	67.8	*****
Fridman et al., 2021 [96]	USA	Linguistic examination	Men with low- and intermediate-risk PCa	n/a	***
Goh et al., 2011 [97]	USA	Cross-sectional survey	AS patients	62	***
Good et al., 2016 [98]	UK	Non-randomized comparative cohort study	AS, BR, EBRT, and RP patients	64.4	****
Gorin et al., 2011 [99]	USA	Prospective survey	AS patients	65.5	**
Guan et al., 2023 [100]	USA	Qualitative interview study	Men with low-risk PCa	61.2	*****
Hegarty et al., 2008 [101]	USA	Retrospective survey	AS patients	USA: 76 Ireland: 76.5	****
Hilger et al., 2019 [102]	Germany	Cross-sectional observational survey study	AS and RP patients	70	***
Hilger et al., 2021 [103]	Germany	Longitudinal survey study	Men with localized PCa	65.5	*****
Hilton et al., 2012 [104]	USA	Prospective cohort study	AS patients	n/a	****
Hirama et al., 2021 [105]	Japan	Prospective cohort survey study	AS patients	n/a	****
Hoffman et al., 2014 [106]	USA	Retrospective cohort study	Men with low-risk PCa	n/a	****
Hoffman et al., 2020 [107]	USA	Prospective population-based cohort study	AS, S, BR, and EBRT patients	n/a	****
Hoffman et al., 2017 [108]	USA	Retrospective cohort survey study	Long-term survivors of men with localized PCa	62	*****
Hoffman et al., 2019 [22]	USA	Prospective cohort study	Men with low-risk PCa	61.5	****
Hoffman et al., 2018 [109]	USA	Cross-sectional survey study	Men with low-risk PCa	n/a	*****
Hogden et al., 2019 [110]	Australia	Qualitative multi-method study with observations, interviews, and surveys	AS patients	n/a	*****
Holmboe & Concato, 2000 [111]	USA	Qualitative interview study	Men with localized PCa	66.4	*****
Huang et al., 2023 [112]	China	Retrospective cohort study	AS and WW patients	64	****
Huber et al., 2017 [113]	Germany	Cross-sectional descriptive survey study	Men with localized PCa	Treatment unchanged: 65 Treatment changed: 66.2	**
Hughes et al., 2022 [114]	UK	Person-based approach with a systematic review, cross-sectional survey, and a qualitative study	AS patients	n/a	***
Huntley et al., 2018 [115]	USA	Prospective survey study	AS patients	n/a	****
Hurwitz et al., 2016 [116]	USA	Prospective cohort study	Men with PCa	61	****
Hurwitz et al., 2017 [117]	USA	Prospective cohort survey study	Men with low- and intermediate-risk PCa	RP: 58 EBRT: 67 BR: 61 AS: 63	****
Isebaert et al., 2008 [118]	Belgium	Qualitative interview study	Men with localized PCa	71	**
Jang et al., 2010 [119]	USA	Retrospective cohort study	Men with localized PCa	n/a	****
Jeldres et al., 2015 [120]	USA	Prospective cohort survey study	AS and RP patients	AS: 65 RP: 58	****
Joseph et al., 2006 [121]	USA	Cross-sectional descriptive study	Men with PCa	70.5	****
Kalapara et al., 2020 [122]	Multi-country	Retrospective cohort study	AS patients	n/a	***
Kan et al., 2018 [123]	USA	Retrospective review study	Men with non-metastatic PCa	n/a	*****
Kang et al., 2022 [124]	USA	Randomized controlled trial	AS patients	63.4	***
Kaye et al., 2018 [125]	USA	Observational cohort study	Men with low-risk PCa	n/a	****
Kayser et al., 2015 [126]	Denmark	Mixed methods based on the Health Literacy Questionnaire	AS patients and their partners	n/a	*****
Kazer et al., 2011 [127]	USA	Single subject internet intervention study	AS patients	70	*****
Kazer et al., 2011 [128]	USA	Qualitative focus group study	AS patients	72	**
Kellogg Parsons et al., 2022 [129]	USA	Secondary analysis of a randomized controlled trial	AS patients	64	***
Kelly et al., 2016 [130]	USA	Retrospective cohort study	AS patients	n/a	*****
Kendel et al., 2016 [131]	Germany	Prospective survey	Men on RP and AS	66.6	***
Kinsella et al., 2019 [18]	UK	Educational seminar intervention	AS patients	Standard: 62.4 Seminar: 63.3	*****
Kirk et al., 2022 [132]	USA	Prospective cohort study	AS patients	63	****
Kord et al., 2023 [133]	USA	Prospective longitudinal cohort study	AS, S, and RT patients	61.6	****
Korman et al., 2013 [134]	USA	Retrospective cohort study	Men with PCa	65.7	**
Lai et al., 2021 [135]	USA	Retrospective cohort study	AS patients	Urology: 73.2 Radiation oncology: 73.5 Medical oncology: 74.5 Primary care: 74.3	*****
Lamers et al., 2017 [136]	The Netherlands	Prospective cohort study	Men with low- and intermediate-risk PCa	65	****
Lamers et al., 2021 [137]	The Netherlands	Randomized controlled trial	Men with low- and intermediate-risk PCa	65.3	****
Lane et al., 2016 [138]	UK	Prospective cohort survey	AS, RP, and RT patients	n/a	**
Lane et al., 2022 [139]	UK	Prospective cohort study within a randomized controlled trial	AM, RP, EBRT, ADT, and BT patients	n/a	***
Lang et al., 2017 [140]	USA	Prospective survey	AS patients	67	****
Latini et al., 2007 [141]	USA	Retrospective survey study	Men with localized PCa	75.5	*****
Le et al., 2016 [142]	USA	Qualitative telephone interviews	Men with localized PCa and their partners	Men: 61.5 Partners: 59.3	*****
Litwin et al., 2002 [143]	USA	Longitudinal survey study	Men with low-risk PCa	65.5	****
Liu et al., 2015 [144]	USA	Retrospective cohort study	Men with PCa	Without AS: 64.5 With AS: 65.4	****
Loeb et al., 2018 [145]	USA	Qualitative study with focus groups and interviews	AS patients	n/a	*****
Loeb et al., 2015 [146]	Sweden	Retrospective cohort study	AS patients and AS providers	n/a	****
Loeb et al., 2013 [147]	Sweden	Retrospective cohort study (population-based study)	Men with low- and intermediate-risk PCa	n/a	****
Lokman et al., 2022 [148]	Finland	Prospective longitudinal cohort study	AS patients	68	***
Luckenbaugh et al., 2022 [149]	USA	Prospective population-based analysis	Men with localized PCa	n/a	*****
Lyons et al., 2016 [150]	USA	Qualitative interview study	Men with low-risk PCa and healthcare providers	65	*****
Mader et al., 2017 [151]	USA	Qualitative interview study	AS patients	65	****
Mallapareddi et al., 2017 [152]	USA	Qualitative focus group study	AS patients and their partners	n/a	*****
Marenghi et al., 2017 [153]	Italy	Prospective cohort study	AS patients	n/a	*****
Martin et al., 2018 [154]	UK	Audit of patients notes	AS patients	65.9	*****
Marzouk et al., 2018 [155]	USA	Prospective cohort study	AS patients	n/a	*****
Matheson et al., 2019 [156]	UK	Mixed methods study with survey and interviews	AS, WW, and AT patients	AS: 68.4 WW: 73.8	***
Matthew et al., 2018 [157]	Canada	Retrospective cross-sectional survey study	AS and RP patients	Age at study AS: 67 RP: 64 Age at treatment AS: 63 RP: 60	*****
Maurice et al., 2015 [158]	USA	Retrospective cohort study	Men with low-risk PCa	n/a	**
McFall et al., 2015 [159]	USA	Mixed methods study using concept mapping approach	Men with localized PCa	n/a	*****
McIntosh et al., 2022 [19]	Australia	Mixed methods study with surveys and interviews	AS patients and patients who discontinued AS	64.5	**
McIntosh et al., 2022 [160]	Australia, USA	Qualitative interview study	Men with localized PCa and their partners	Men: 59.6 Partners: 59.9	*****
Menichetti et al., 2019 [161]	Italy	Qualitative focus group study	AS patients	68	*****
Merriel et al., 2019 [162]	UK	Consensus statement based on systematic review, survey, and interviews	AS patients	n/a	***
Mills et al., 2006 [163]	UK	Randomized controlled trial	Men with localized PCa	62.5	***
Mishra et al., 2013 [164]	USA	Content analysis of online patient conversations	Internet conversations regarding PCa treatment	n/a	*****
Monaco et al., 2022 [165]	USA	Prospective cohort study	Men with low-and intermediate-risk PCa	66	*****
Moon et al., 2019 [166]	USA	Population-based prospective cohort study	SBRT, EBRT, and AS patients	AS: 66 EBRT: 66 SBRT: 65	*****
Mroz et al., 2013 [167]	Canada	Qualitative interview study	Men with low-risk PCa	68	*****
Myers et al., 2018 [168]	USA	Pilot study; prospective survey study	Men with low-risk PCa	n/a	****
Naha et al., 2021 [169]	USA	Retrospective cohort study	AS patients	n/a	*****
Nguyen-Nielsen et al., 2020 [170]	Denmark	Longitudinal cohort study	Men with PCa	n/a	*****
Nielsen et al., 2020 [171]	Denmark	Qualitative interview study	AS patients	n/a	*****
Nilsson et al., 2021 [172]	Norway	Cross-sectional study	AS and RP patients	At diagnosis: 60.9 At survey: 65	*****
O’Callaghan et al., 2014 [173]	Australia	Qualitative interviews	Men with low-risk PCa and their partners	n/a	*****
Oliffe et al., 2009 [174]	Canada	Qualitative interview study	AS patients	68	*****
Olsson et al., 2020 [175]	Sweden	Population-based cohort study	AS and AT patients	n/a	****
Orom et al., 2014 [176]	USA	Cross-sectional survey	AS patients	64.7	***
Orom et al., 2017 [177]	USA	Prospective survey	Patients undergoing AS, RT, and RP	n/a	*****
Otto et al., 2022 [178]	Germany	Longitudinal study	Men with localized PCa	65.8	****
Papadopoulos et al., 2019 [179]	Canada	Prospective cohort study	Men who discontinued AS	61.8	*****
Parikh et al., 2017 [180]	USA	Retrospective cohort study	Patients with low-risk PCa	AS: 63.3 Curative intervention: 61.8	*****
Parker et al., 2016 [181]	USA	Prospective cohort survey	AS patients	67.2	****
Paudel et al., 2021 [182]	USA	Retrospective study	Men with localized PCa	n/a	****
Pearce et al., 2015 [183]	USA	Prospective longitudinal study	AS patients	66.5	****
Pham et al., 2016 [184]	USA	Prospective survey study	Men who underwent prostate needle biopsy	AS: 64 No cancer: 61	*****
Pozzar et al., 2022 [185]	USA	Prospective survey study within a multicenter RCT	Men with localized PCa with AS, S, and RT treatment	n/a	**
Punnen et al., 2013 [186]	USA	Prospective cohort survey	AS and RP patients	60.5	****
Radhakrishnan et al., 2018 [187]	USA	Retrospective survey study	Men with newly diagnosed localized PCa	65	*****
Reamer et al., 2017 [188]	USA	Population-based cross-sectional survey study	WW, AS, S, and RT patients	61, AS: 64.6	*****
Remmers et al., 2023 [189]	The Netherlands	Retrospective patient-reported outcome study across data sources	AS, RP, and RT patients	n/a	****
Repetto et al., 2016 [190]	Italy	Prospective survey study	Men who discontinued AS	71	*****
Richard et al., 2016 [191]	Canada	Retrospective population-based study	AS, WW, and AT patients	n/a	*****
Rossen et al., 2016 [192]	Denmark	Semi-structured interview study	Spouses of men with early-stage PCa	n/a	*****
Ruane-McAteer et al., 2019 [193]	UK	Longitudinal cohort study	Men with newly diagnosed localized PCa	AS: 64.9 AT: 62.2 No PCa: 61.8	****
Scherr et al., 2017 [194]	USA	Prospective cohort study	Men with localized PCa	Patients: 63.2	*****
Sciarra et al., 2018 [195]	Italy	Single-center prospective non-randomized survey study	RP, EBRT, and AS patients	AS: 70.84 RT: 70.63 RP: 65.34	*****
Seaman et al., 2019 [196]	USA	Semi-structured interviews	Men with low-risk PCa	70.4	*****
Seiler et al., 2012 [197]	Switzerland	Prospective cross-sectional survey study	AS patients and their partners	69.3	****
Shankar et al., 2019 [198]	USA	Prospective observational patient-reported outcome study	Men scheduled for mpMRI or transrectal prostate biopsy	n/a	****
Shelton et al., 2019 [199]	USA	Retrospective chart review	AS patients	n/a	*****
Sidana et al., 2012 [200]	USA	Retrospective survey study	RT, AS, and S patients	S: 45.7 RT: 46.6 AS: 46.6 Other: 42.9	****
Smith et al., 2009 [201]	Australia	Population based cohort study	Men with localized PCa	61.2	*****
Sureda et al., 2019 [202]	France	Retrospective cross-sectional survey study	AS, EBRT, RP, and BT patients	AS: 70.6 RP: 68.2 EBRT: 73.2 BR: 71.9	*****
Sypre et al., 2022 [203]	USA	Prospective interview study	RP, RT, and AS patients	70.9	****
Tan et al., 2016 [204]	USA	Prospective survey study	AS patients	n/a	*****
Taylor et al., 2016 [205]	USA	Longitudinal cohort study with telephone interviews	Men newly diagnosed with low-risk PCa	61.46	*****
Taylor et al., 2018 [2]	France	Prospective interview study	Men with newly diagnosed Pca	n/a	*****
Teunissen et al., 2023 [206]	The Netherlands	Survey study with preference elicitation	Men with localized Pca and healthy volunteers	n/a	***
Thurtle et al., 2021 [207]	UK	Multicenter RCT	Men with newly diagnosed PCa	n/a	****
Timilshina et al., 2021 [208]	Canada	Observational population-based study	AS, WW, and AT patients	AS: 63.9 In treatment: 62.4 WW: 72.9	****
Tiruye et al., 2023 [209]	Australia	Prospective survey study	AS patients	65	*****
Tiruye et al., 2022 [210]	Australia	Prospective survey study	AS, RP, EBRT, and BT patients	66.1	*****
Todio et al., 2023 [211]	Australia	Semi-structured interviews	AS, BT, EBRT, RP, and NanoKnife patients	65.65	*****
Tohi et al., 2020 [212]	Japan	Retrospective analysis on a prospective cohort	AS patients	n/a	****
Tohi et al., 2022 [213]	Japan	Prospective longitudinal survey study	AS patients	n/a	****
van den Bergh et al., 2010 [214]	The Netherlands	Prospective cohort survey study	AS patients	n/a	*****
van den Bergh et al., 2009 [12]	The Netherlands	Prospective cohort study	AS patients	64.9	*****
van den Bergh et al., 2012 [215]	The Netherlands	Non-randomized, comparative cohort study	AS, RP, and RT patients	AS: 64.9 RP: 62.1 RT: 68.1	*****
van den Bergh et al., 2010 [216]	The Netherlands	Prospective survey study	AS patients	n/a	****
Van Hemelrijck et al., 2019 [16]	Multi-country	Retrospective cohort study	AS patients	65	****
van Stam et al., 2020 [217]	The Netherlands	Prospective observational multicenter study	RP, EBRT, BT, and AS patients	66.4	*****
van Stam et al., 2018 [218]	The Netherlands	Prospective longitudinal survey study	Men with newly diagnosed localized PCa	66.46	****
van Vugt et al., 2011 [219]	The Netherlands	Prospective survey study	Men with PCa, their urologists	64	*****
Vanagas et al., 2013 [220]	Lithuania	Prospective survey study	AS, S, RT, HT, chemotherapy, and combined treatment patients	64	****
Vasarainen et al., 2012 [221]	Finland	Prospective cohort study	AS patients	n/a	*****
Venderbos et al., 2017 [25]	The Netherlands	Retrospective survey on HRQoL	AS, RP, and RT patients	AS: 65.3 RP: 70 RT: 65.9	*****
Venderbos et al., 2022 [222]	The Netherlands	Cross-sectional survey study	AS, RP, and RT patients	n/a	***
Venderbos et al., 2023 [223]	Multi-country	Cross-sectional survey study	Men undergoing treatment for PCa or previous PCa treatment	n/a	***
Venderbos et al., 2015 [224]	The Netherlands	Retrospective survey on anxiety and distress levels	AS patients	64.6	****
Volk et al., 2015 [225]	USA	Qualitative survey study	Men with localized PCa	64.9	****
Volk et al., 2014 [226]	USA	Qualitative interviews	AS, RP, and RT patients	AS: 62.6 AT: 58.6	*****
Vos et al., 2018 [227]	Canada	Retrospective cohort study	RP, RT, HT, and AS patients	n/a	****
Wade et al., 2020 [228]	UK	Longitudinal qualitative interview study	Men with localized PCa	n/a	*****
Wade et al., 2015 [229,230]	UK	Qualitative interview study	AS patients, urologists, and nurses	AS: 65 AM: 64.7	***
Wade et al., 2015 [230]	UK	Qualitative interview study	AS patients, urologists, and nurses	63.6	*****
Wade et al., 2013 [231]	UK	Prospective cohort survey study	Men undergoing a prostate biopsy	Negative biopsy: 62 Cancer: 62.3	****
Wadhwa et al., 2017 [232]	UK	Prospective cohort survey study	Men undergoing a prostate biopsy	n/a	****
Wagland et al., 2019 [233]	UK	Cross-sectional survey with semi-structured interviews	Men with recent PCa diagnosis	Survey: 71.09 Interview: 65.5	****
Walker & Santos- Iglesias, 2023 [234]	Canada	Prospective cohort survey study	Men who underwent a biopsy, and men who did not undergo a biopsy and have no PCa	AS: 62.25 CG: 58.76 Neg. B: 60.8 AT: 61.95	***
Wallis et al., 2022 [235]	Canada, USA	Prospective cohort study	S, RT, and AS patients	n/a	*****
Watson et al., 2016 [236]	UK	Questionnaire	AS, S, RT, and HT patients	n/a	*****
Watts et al., 2015 [237]	UK	Cross-sectional survey	AS patients	70.49	*****
Weerakoon et al., 2015 [238]	Australia	Retrospective registry study	AS patients	n/a	****
Wilcox et al., 2014 [239]	Australia	Prospective survey	AS patients	62	***
Womble et al., 2015 [240]	USA	Retrospective cohort study	Men with localized PCa	n/a	*****
Xu et al., 2011 [241]	USA	Qualitative interviews	Men with localized PCa	64.3	*****
Xu et al., 2016 [242]	USA	Cross-sectional survey	Men with localized PCa	S: 59.4 RT: 63.2 AS/WW: 64.3	*****
Xu et al., 2016 [243]	USA	Cross-sectional survey	Men with localized PCa	61.2	****
Xu et al., 2012 [244]	USA	Qualitative interviews	Men with low-risk PCa	58	****
Yanez et al., 2015 [245]	USA	Cross-sectional study	AS patients	64.40	*****
Zeliadt et al., 2010 [246]	USA	Multisite survey	Men with localized PCa	63	*****

Abbreviations: AS, active surveillance; AT, active treatment; ADT, androgen deprivation therapy; BR, brachytherapy; C, chemotherapy; CRT, conformal external beam radiotherapy; EBRT, external beam radiotherapy; HT, hormonal therapy; IP, individual practitioners; MC, multidisciplinary clinic; MMAT, mixed methods appraisal tool; RP, radical prostatectomy; RT, radiotherapy; S, surgery; SBRT, stereotactic body radiotherapy. * Very low, ** Low, *** Moderate, **** High, ***** Very high.

**Table 2 jpm-15-00315-t002:** Summary table of factors influencing choice, adherence to and the discontinuation of active surveillance among men with low- and intermediate-risk prostate cancer found in the systematic review (in alphabetical order).

Level	Factor	References
Clinical level	Cancer characteristics: risk, PSA levels, grading, staging, and tumor volume	[16,19,36,42,60,93,125,132,146,151,152,153,162,175,199,208,226,238,241]
	Healthcare providers acceptance of AS	[77,93,106,144,164,167,196]
	Consultations: physician recommendation, shared decision-making, language used during consultations, time between diagnosis and treatment decision, and decision aids	[22,43,44,51,53,72,74,75,76,84,87,91,92,94,95,96,97,99,100,109,111,118,136,137,150,152,159,160,167,168,173,176,177,185,187,188,194,200,205,206,207,211,225,226,233,241,243,244,246]
	Specialty of healthcare provider	[45,50,65,67,68,106,119,135,154,188,229]
	Healthcare provider–patient relationship: trust	[48,150,151,152,167,173,176,196,226]
	Type of healthcare institution: multidisciplinary clinic [22,43,44,51,53,72,74,75,76,84,87,91,92,94,95,96,97,99,100,109,111,118,136,137,150,152,159,160,167,168,173,176,177,185,187,188,194,200,205,206,207,211,225,226,233,241,243,244,246], academic hospital, and regional differences	[32,45,48,68,88,116,123,134,158,180,227,238]
Patient level	Avoiding side-effects of active treatment (sexual function, urinary problems)	[36,42,51,57,59,71,75,76,77,99,100,150,151,160,182,196,219,226]
	Impact during AS: prostate biopsy burden, sexual and urinary function, QoL, and HRQoL	[[9],[33],[39],[41],[45],[47],[52],[56],[58],[61],[64],[65],[69],[70],[71],[73],[75],[79],[80],[81],[82],[85],[93],[94],[98],[101],[102],[104],[105],[107],[115],[120],[122],[133],[138],[139],[148],[151],[152],[161],[165],[166],[170],[173],[174],[181],[183],[184],[189],[195],[198],[201],[202],,[209],[210],[212],[213],[215],[217],[220],[221],,[222],[223],[230],[231],[232],[234],[236],[239],[240]]
	Perception of cancer risk: fear of progression, fear of reoccurrence	[19,37,52,83,110,131,159,171,172,218,242,244]
	Knowledge about AS: seeking information, decision regret during treatment	[33,43,44,48,49,54,55,66,79,90,94,98,103,108,117,128,145,150,162,190,196,216,217,235,244]
	Interventions during AS: diet, exercise	[18,73,78,114,124,127,129,179,229]
	Patient characteristics: age, race, comorbidities, education, family history, socioeconomic status, and marital status	[32,34,38,46,59,62,75,76,112,130,144,146,147,158,163,175,180,191,205,233]
	Psychological impact: stress, anxiety, depression, and uncertainty	[[12],[35],[37],[40],[44],[60],[79],[83],[86],[89],[121],[140],[141],[143],[149],[153],[155],[156][157],[163],[164],[169],[171],[173],[174],[178],[181],[186],[193],[195],[197],[203],[204],[214],[219],[224],[228],[230],[231],[237],[239],[245]]
	Religion and spirituality	[33,40,100,159]
	Knowledge about PCa in family and friends: fear of progression [12,35,37,40,44,60,79,83,86,89,121,140,141,143,149,153,155,156,157,163,164,169,171,173,174,178,181,186,193,195,197,203,204,214,219,224,228,230,231,237,239,245]	[142,152,164]
Social level	Social pressure: family encouragement	[44,53,140,174,192]
	Social support from partner/children/family/friends/others; family history with PCa	[33,40,48,52,89,100,110,126,128,142,151,160,171,192]
	Support needs: support groups, advice from peers	[40,44,63,113,145,173]

## Data Availability

No new data were created or analyzed in this study. Data sharing is not applicable to this article.

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
