# Peer review of "Key Determinants Influencing Treatment Decision-Making for and Adherence to Active Surveillance for Prostate Cancer: A Systematic Review"

_jpm, 2025, doi:10.3390/jpm15070315_

Round 1
Reviewer 1 Report
Comments and Suggestions for Authors
Congratulations for your paper. Impressive work. As you wrote, the studies are very heterogenous; maybe you should be more specific at the conclusions
I think the topic is relevant. Active surveillance (AS) is increasingly recommended as a first-line management strategy for patients with low-risk or favorable intermediate-risk prostate cancer.
The review is based on the evaluation of a very large number of articles (223) and that is why I think is addressing a gap in the field. The number of articles also explains the references list.
I think the conclusions should be more consistent/detailed and focused on the objective of this review.
Reviewer 2 Report
Comments and Suggestions for Authors
Comments regarding the review article titled: Key determinants influencing treatment decision-making for and adherence to active surveillance for prostate cancer: A systematic review
The authors made a thorough search of the literature available regarding prostate cancer and active surveillance in various databases. They used the PRISMA guidelines to perform the article selection and evaluated the confidence of each article using MMAT which provides an additional layer of robustness to this review. The main goal of the authors was to identify the most relevant factors related to adherence and treatment decision making regarding active surveillance. The article is very relevant since active surveillance is one of the treatment options for low-risk prostate cancer and it is extremely important to understand the social, medical, and patient-related issues that reduce adherence in prostate cancer patients.
A few comments:
The resolution of the figures should be improved to increase readability.
It would be interesting if the authors could include a percentage of the studies that addressed a specific factor in Figure 2. It would provide a better picture of the factors with the greatest influence on each category.
Reviewer 3 Report
Comments and Suggestions for Authors
This manuscript presents a systematic review aiming to identify the factors that influence treatment decision-making (TDM) and adherence to active surveillance (AS) among men with low- and intermediate-risk prostate cancer. It is a highly relevant and timely contribution, particularly given the growing emphasis on shared decision-making and quality-of-life preservation in urologic oncology.
The paper is methodologically rigorous, PRISMA-compliant, and includes a vast number of studies (n = 223), making it an authoritative reference on the subject. However, some improvements in structure, clarity, and critical synthesis would significantly enhance its scientific impact and reader accessibility.
Major Strengths
-
Comprehensive Scope:
The inclusion of 223 studies across clinical, patient-related, and social dimensions demonstrates excellent breadth. -
Methodological Rigor:
The authors followed the PRISMA guidelines and used a pre-registered protocol (PROSPERO), adding transparency and reproducibility to the review. -
Categorization of Factors:
The use of a multilevel framework (clinical, patient, social) is logical and improves clarity in navigating complex determinants. -
Quality Assessment:
Use of the MMAT tool is appropriate. Reporting quality scores for each study lends credibility to the findings.
Major Suggestions for Improvement
1. Overly Descriptive Results
-
The Results section often reads like a long list of study findings. More critical synthesis is needed, especially by:
-
Aggregating evidence (e.g., “X% of studies identified fear of progression as a key factor”).
-
Identifying trends (e.g., by time, geography, or study design).
-
Discussing conflicting evidence.
-
Suggestion: Use summary tables/figures more effectively and integrate meta-analytic commentary where applicable.
2. Clarity and Language
-
The manuscript is generally well-written, but some sections are dense and repetitive, especially where individual studies are cited back-to-back without synthesis.
-
Occasional awkward or overly informal phrases occur, such as:
-
“AS is described as having ‘no treatment’” → consider “AS is often misunderstood by patients and families as withholding treatment.”
-
Suggestion: Consider light language editing to improve conciseness and flow.
3. Lack of Emphasis on Clinical Implications
-
The Discussion would benefit from greater emphasis on practical recommendations for clinicians, such as:
-
How to frame risk communication more effectively.
-
How to incorporate family in decision-making.
-
What psychosocial support structures are most impactful?
-
Suggestion: Add a dedicated subsection on implications for clinical practice and/or policy.
4. Missing Limitations and Biases
-
The review lacks a thorough limitations section. For example:
-
Possible publication bias.
-
Lack of standardization in defining “adherence” across studies.
-
Inclusion limited to English-language publications.
-
Suggestion: Include a paragraph on limitations in the Discussion.
Minor Comments
-
Abstract: Clearly written but may benefit from including the number of included studies and databases searched.
-
Figures/Tables: Figure 2 is valuable but could be refined graphically for better readability.
-
Redundancy: Several points about anxiety, biopsy burden, and social support are repeated in both the decision-making and adherence sections.
- References: COnsider this reference for PCa and genomic (doi: 10.3390/cancers16223766.) and cosider this related to alterantive treatment for PCa (doi: 10.3390/jcm13092551.)
